# *Trichoderma brevicompactum* 6311: Prevention and Control of *Phytophthora capsici* and Its Growth-Promoting Effect

**DOI:** 10.3390/jof11020105

**Published:** 2025-01-30

**Authors:** Jien Zhou, Junfeng Liang, Xueyan Zhang, Feng Wang, Zheng Qu, Tongguo Gao, Yanpo Yao, Yanli Luo

**Affiliations:** 1College of Resources and Environment, Xinjiang Agricultural University, Urumqi 830052, China; zjientongxue@163.com; 2Agro-Environmental Protection Institute, Ministry of Agriculture and Rural Affairs, Tianjin 300191, China; liangjunfeng@caas.cn (J.L.); 13571732795@163.com (X.Z.); wangfeng_530@163.com (F.W.); quzhengwf@163.com (Z.Q.); 3College of Life Sciences, Hebei Agricultural University, Baoding 071002, China; gtgrxf@163.com

**Keywords:** *T. brevicompactum*, biocontrol strains, soil-borne pathogens, pepper Phytophthora blight, disease inhibition, growth promotion

## Abstract

Pepper Phytophthora blight caused by *Phytophthora capsici* results in substantial losses in global pepper cultivation. The use of biocontrol agents with the dual functions of disease suppression and crop growth promotion is a green and sustainable way of managing this pathogen. In this study, six biocontrol strains of *Trichoderma* with high antagonistic activity against *P. capsici* were isolated and screened from the rhizosphere soil of healthy peppers undergoing long-term continuous cultivation. Morphological identification and molecular biological identification revealed that strains 2213 and 2221 were *T. harzianum*, strains 5111, 6311, and 6321 were *T. brevicompactum*, and strain 7111 was *T. virens*. The results showed that *T. brevicompactum* 6311 had the greatest inhibitory effect against *P. capsici*. The inhibition rate of 6311 on the mycelial growth of *P. capsici* was 82.22% in a double-culture test, whereas it reached 100% in a fermentation liquid culture test. Meanwhile, the pepper fruit tests showed that 6311 was 29% effective against *P. capsici* on pepper, and a potting test demonstrated that the preventive and controlling effect of 6311 on pepper epidemics triggered by *P. capsici* was 55.56%. The growth-promoting effect, germination potential, germination rate, radicle-embryonic axis length, germination index, and fresh weight of peppers cultured in the 6311 fermentation broth were significantly increased compared with the results for the control group. Scanning electron microscopy revealed that 6311 achieved the parasitism of *P. capsici*, producing siderophores and the growth hormone indoleacetic acid (IAA) to achieve disease-suppressive and growth-promoting functions. Transcriptomic results indicated that genes encoding proteins involved in plant disease resistance, namely flavanone 3-hydroxylase (F3H) and growth transcription factor (AUX22), were generally upregulated after the application of 6311. This study demonstrated that 6311 exhibits significant bioprotective and growth-promoting functions.

## 1. Introduction

Peppers (*Capsicum annuum* L.) are vegetables that are well known and widely consumed globally, and they are prized for their culinary versatility and unique flavor. Yearly accumulation of pathogenic fungi in the soil due to continuous cropping can lead to outbreaks of soil-borne diseases [1,2]. As a result of intensive and protected cultivation, soil-borne diseases are occurring extensively. Pepper yields have been reduced by more than 60%, causing significant economic losses to farmers and seriously hampering the development of the pepper industry [3,4].

Cultivated peppers are highly susceptible to *P. capsici*. Pepper Phytophthora blight caused by this pathogen tends to develop rapidly, resulting in severe yield loss or even crop failure. *P. capsici* is a fungal pathogen capable of causing rot and wilt in a wide range of crops. The sporangia of *P. capsici* can germinate and invade plant cells directly or produce zoospores, which are released into water, attaching to plant roots and producing germ tubes that penetrate plant cells to cause pepper Phytophthora blight. This disease is one of the most common soil-borne fungal diseases globally and can occur at all reproductive stages of peppers [5]; thus, soil contaminated by *P. capsici* is the main cause of pepper Phytophthora blight. The application of chemical pesticides is currently the most common method for managing pepper Phytophthora blight; however, the long-term application of pesticides not only destroys the health of the soil environment, but also tends to cause pepper Phytophthora blight to become mold resistant. For this reason, adopting biocontrol measures that are both environmentally friendly and crop safe would be a very promising strategy. Therefore, the application of functional microorganisms and their metabolites is proposed as an alternative to conventional chemical pesticides.

Soil microbial communities are essential components of soil ecosystems and are important indicators of soil microbiological health [6]. The fungi *Trichoderma* spp., living in soil, can inhibit the growth of *P. capsici*, improve plant resistance to pathogens, and promote plant growth, regulating the microbiological structure of the soil. It has been successfully developed for use as a commercial biocontrol agent [7,8,9,10], such as in the form of *T. harzianum*, *T. hamatum*, *T. virens*, and *T. asperellum* [11]. These species are effective against a wide range of pathogens, such as *Phytophthora*, *Fusarium*, and *Pythium* [12]. *Trichoderma* spp. antagonizes pathogens through a variety of direct and indirect mechanisms. The direct mechanisms of their biocontrol include competition for space and nutrients [12], as well as secretion of lytic enzymes [13] and antibiotics [14]. For example, *T. asperellum* can surround and penetrate *P. capsici* hyphae, leading to mycelial collapse, and can also use spores to enter *P. capsici* oospores, develop mycelium, and produce conidia, thereby leading to the decomposition of *P. capsici* oospores [15]. Meanwhile, the antibiotic gliadin produced by *T. virens* has been shown to inhibit the growth of *Mycosphaerella* spp., with a significant reduction (by 62.64%) in disease incidence [16]. The main indirect mechanisms of their biocontrol include the induction of plant resistance, increased resistance to stress, and root growth. For example, lilac seedlings treated with *T. afroharzianum* T52 biofertilizer showed a significant increase in the number of lateral roots, and the incidence of the plant infection was reduced by 12.28% compared with that in the control [17]. Meanwhile, *T. ghanense* and *T. citrinoviride* increased the shoot length, root length, leaf length, leaf width, and dry weight of cucumbers, while suppressing disease caused by *P. infestans* [18]. The application of *T. viride* not only significantly reduced the disease index of soybeans, but also favored the maintenance of network stability and increased network complexity, as well as being more friendly to soil microhabitats [19]. *T. brevicompactum* controlled root rot of *Atractylodes macrocephala* caused by *Fusarium oxysporum*, reducing the severity of root rot by 48% to 58% and significantly increasing plant growth and biomass compared to those of the control (healthy) group [20].

Peppers from Jize County, Handan City, Hebei Province, China, were used as court tributes during the Ming and Qing dynasties, while Wangdu County in Baoding also has a 600-year history of pepper cultivation. In this study, healthy pepper plants were collected from soil with long-term continuous pepper cropping in Hebei Province, China, to perform targeted cultivation and screening of rhizosphere soil strains, along with the isolation and identification of functional strains. The biocontrol and biopromotion performance and mechanism of action of biocontrol microorganisms against *P. capsici* were verified through in vitro tests, such as the double-culture test, the fermentation liquid culture test, and the isolated peppers fruit test, along with in vivo tests, such as the seed germination test and the pot planting test. The goals of this study will provide insights into the effective prevention and control of pepper Phytophthora blight and promote the sustainable production of peppers.

## 2. Materials and Methods

### 2.1. Source, Isolation, and Identification of Biocontrol Fungi

Source of biocontrol fungi and pathogenic strains: From July to September 2023, blight occurred in some long-term continuous pepper cropping plots in Handan City and Baoding City, Hebei Province, P.R. China, resulting in massive plant deaths. Two batches of inter-root soil of healthy pepper plants were collected from long-term continuous cropping plots during the blight outbreaks and stored at 4 °C. The collected soil was used to cultivate potential biocontrol fungi and isolate and purify potential biocontrol fungi. *P. capsici* was provided by the Institute of Plant Protection, Hebei Academy of Agricultural and Forestry Sciences, Hebei Province, China.

Isolation and screening of biocontrol fungi: A total of 5 g of fresh inter-root soil was placed in 45 mL of sterile water and shaken on a shaker at 28 °C and 180 rpm for 1 h. The soil samples were diluted to 10^−1^, 10^−2^, 10^−3^, 10^−4^, 10^−5^, and 10^−6^ using the stepwise dilution method, and 100 µL of each soil suspension was aspirated on potato dextrose agar (PDA) medium plates for spreading; this procedure was repeated three times. After incubation at 28 °C for 3 days, single colonies were picked up with an inoculation ring and transferred to a PDA plate for further purification. This step was repeated five times until pure colonies were obtained, which were numbered and stored in a refrigerator at 4 °C.

Biocultural testing of biocontrol fungi and *P. capsici*: A 5 mm radius sterilized hole punch was used to punch holes at the edge of the *P. capsici* colonies cultured for 5 days. The *P. capsici* section was inoculated on one side of a 9 cm diameter PDA plate, and the symmetrical position of the *P. capsici* block was inoculated with the six potential biomicrobial blocks to create a 5 cm distance between the blocks. The plate was then sealed with sealing film and subjected to inverted incubation for 7 days in the dark at 28 °C. Only the PDA plate inoculated with the *P. capsici* discs was used as a control, and each treatment was replicated three times. The percentage of fungi inhibition was calculated using the crossover method (see Equation (1)).(1)$$\mathrm{Inhibition}\;\mathrm{rate}=\frac{\left(D-d\right)}{D}\;\times \;100\%$$

D: diameter of *P. capsici* in the control group; d: diameter of *P. capsici* in the treatment group

Identification of biocontrol microbial strains: (1) The strains to be identified were inoculated onto PDA plates and cultured at a constant temperature of 28 °C. After 5 days, the morphology and color of the colonies were observed, and the morphological characteristics of conidia and conidiophores were observed under a light microscope.

Molecular biological identification: The cultured strains were ground to fine powder in liquid nitrogen, and total genomic DNA was extracted from mycelium using a modified version of the CTAB method described by Tiwari [21]. The following primers were used: rDNA internal transcribed spacer region ITS4: (5′-TCCTCCGCTTATTGATATATGC-3′); ITS5: (5′-GGAAGTAAAAGTCGTAACAAGG-3′). PCR was performed in accordance with the procedure of Korkom [22]. The ITS sequences of the biocontrol microorganisms were constructed, along with the sequences of other *Trichoderma* from the GenBank database of NCBI, and the phylogenetic tree was constructed using the neighbor-joining method in the MEGA11 program. The ITS sequences of the biocontrol microorganisms have been submitted to the GenBank database of NCBI. Further confirmation was made through *tef1* sequencing, which can distinguish different strains of Trichoderma, using the primer pair EF1-728F and TEFl rev and PCR conditions as described by Samuels [23].

### 2.2. Inhibition of the Disease and Enhancement of Growth Performance by the Biocontrol Organisms

In vitro inhibitory activity testing of fermentation solution against *P. capsici*: Activated PDA plates of six *Trichoderma* strains developed on potato dextrose liquid medium (PDB) were inoculated conical and cultured by oscillation at 180 rpm and 28 °C for 7 days. The fermentation broth was obtained by filtration through eight layers of sterile gauze and then a 0.22-μm sterile filter membrane. The plates were made by mixing the fermentation broth with PDA at a ratio of 1:3 (by volume), and the center was inoculated with fresh cakes of *P. capsici*, while equal-volume PDA plates, without the addition of the fermentation broth, were used as a control. Each treatment was repeated three times. The other procedures were the same as those described in Section 2.1.

Fermentation solution for the pepper promotion test: Healthy, uniform-sized pepper seeds were disinfected with 5% NaClO for 5 min, rinsed well with sterile water, drained, and placed in a five-fold dilution of fermentation solution of six *Trichoderma* strains for 24 h. The seeds were evenly placed in a germination box with a double layer of aseptic germination paper, 50 seeds per dish, and each treatment was replicated three times, followed by culture in a humidified atmosphere at 28 °C for 10 days. Sterile water was used for a control group to determine the germination potential of the seeds, germination rate, fresh weight, embryonic axes’ radicle length, and 5-day germination index.


Germination rate (%) = number of seeds germinated at 10 days/total number of seeds(2)
Germination potential (%) = number of seeds germinated at 5 days/total number of seeds × 100%(3)

(4)
$$\mathrm{Germination}\;\mathrm{index}\;(\mathrm{GI})=\sum Gt\frac{Gt}{Dt}$$



Gt is the number of germinations per day, and *Dt* is the number of germination days.

Pepper fruit blight antagonism test: Peppers that were disease-free, unwounded, mature, and basically the same size were selected and immersed in 2% sodium hypochlorite solution for 5 min, followed by rinsing with sterile water three times, and then left to dry. A wound of 5 mm in diameter and 1 mm in depth was created at the waist of the pepper using a sterile perforator, and six discs of six strains of biocontrol microorganisms were inoculated into the wound. The wounds were covered with blotting paper that absorbed a sufficient amount of sterile water, and the fruits were placed in sterilized plastic pots covered with film to keep them warm and moisturized. After 24 h of incubation at room temperature, the fungal cake was removed, and then blocks of *P. capsici* were inoculated into the wounds to continue incubation. The diseased area was observed and photographed every day, and the area was determined, with each treatment being repeated three times. Treatment involving inoculation with *P. capsici* alone was used as a positive control, while treatment with sterile water was used as a negative CK. The inhibition rate was calculated as described in Section 2.1.

Potting experiment: Six treatments were designed for potting. Treatment 1 was the sterile water control (CK), treatment 2 was *Trichoderma* 2213 + *P. capsici*, treatment 3 was *Trichoderma* 6311 + *P. capsici*, treatment 4 was *P. capsici*, treatment 5 was *Trichoderma* 2213, and treatment 6 was *Trichoderma* 6311.

Experimental method: Three peppers of TianYu No. 5 were planted in each pot, cultivated for 90 days, followed by wounding of the roots, and in treatments 2, 3, 5, and 6, respectively, 10 mL of *Trichoderma* spore solution with a spore count of 1 × 10^6^ was added to the pepper’ root systems. Five days after inoculation with the *Trichoderma* suspensions, a suspension of 1 × 10^5^ conidia of *P. capsici* was added in treatments 2, 3, and 4, with each treatment being replicated three times. All roots of the plants were collected on the 5th day of inoculation of peppers with *P. capsici*. Disease severity was rated as follows: level 0: the whole plant was free of disease; level 1: less than one-fifth of the lateral branches were affected, or watery spots appeared at the base of the stems; level 2: one-fifth to one-half of the lateral branches were affected, or decay spots appeared at the base of the stems, but the plants had not wilted; level 3: one-half to three-quarters of the lateral branches were affected, or decay spots appeared at the base of the stems, and the plants showed recoverable wilting; level 4: more than three-quarters of the whole plant’s lateral branches were affected, some branches were dead, or the plant showed irrecoverable wilting phenomenon; and level 5: the whole plant was dead.(5)$$\mathrm{Blight}\;\mathrm{disease}\;\mathrm{condition}\;\mathrm{index}=\frac{\Sigma Disease\;severity\;of\;diseased\;plants\times number\;of\;diseased\;plants}{Highest\;incidence\;severity\;\times \;total\;number\;of\;plants\;surveyed}$$(6)$$\mathrm{Prevention}\;\mathrm{and}\;\mathrm{control}\;\mathrm{effects}=\frac{Comparison\;Condition\;Index-Treatment\;Condition\;Index}{Comparison\;of\;condition\;indices}$$

### 2.3. Analysis of Disease-Suppressing and Growth-Promoting Mechanisms of Biocontrol Microorganisms

Comparison of antagonistic properties between strains: Scanning electron microscopy (SEM) was used to examine the role of biocontrol fungi on the plates in the double-culture experiment. The above junction isolated the strains of biocontrol fungi, and *P. capsici* was cut off with a sterile blade, observed, and recorded using a light microscope. The junction was then collected and fixed in a 1.5 mL centrifuge tube, 1 mL of 2.5% glutaraldehyde solution was added, and the samples were rinsed with 0.1 M, pH 7.0 phosphate buffer three times for 15 min each. The samples were fixed with 1% osmioglycolic acid solution for 1–2 h and rinsed with 0.1 M, pH 7.0 phosphate buffer three times for 15 min each. The samples were then dehydrated with five solutions containing increasing concentrations of ethanol. The samples were rinsed three times with 0.1 M, pH 7.0 phosphate buffer solution for 15 min each. Next, the samples were dehydrated with five solutions containing increasing concentrations of ethanol for 15 min and then treated with 100% ethanol twice for 20 min each. The samples were treated with a mixture of ethanol and isoamyl acetate (1:1 by volume) for 30 min and then treated with pure isoamyl acetate for 1 h. Finally, the samples were subjected to critical point drying, coated, and then observed via SEM.

Siderophores performance test: Six strains of the potential biocontrol microbial were cultured on PDA plates for 7 days to achieve colony growth. Then, the cooled Chrome azurol S (CAS) test medium was poured onto the PDA plates and left to stand for 24 h to observe the plate color. Each biocontrol microbial strain comprised one treatment, and each treatment was repeated three times. The spore liquid was inoculated into PDB with 5% (*v*/*v*) inoculum, incubated at 28 °C and 180 rpm for 5 days, and then centrifuged at 8000 rpm for 15 min to obtain the supernatant. A total of 3 mL of CAS assay solution was added to the polycarbonate tube, and then 3 mL of supernatant was added, followed by mixing, and the mixture was allowed to stand for 1 h. Absorbance was detected using a spectrophotometer at 630 nm (A1). A total of 3 mL of uninoculated PDB and 3 mL of supernatant were removed to observe the color of the plate, with each treatment being repeated three times. The inoculated PDB and 3 mL of CAS assay solution were mixed well, and the absorbance was measured in the same way as for the reference value (A0). The rate of synthesis of the siderophores was determined using Equation (7).(7)$$\mathrm{Relative}\;\mathrm{concentration}\;\mathrm{of}\;\mathrm{siderophores}\;(\%)=\frac{\left(A0-A1\right)}{A0}\;$$

IAA growth-promoting properties: We referred to the method used in the research of Ma et al. and improved upon it [24]. The bioprophylactic microorganisms were activated on PDA plates at 28 °C for 5 days. Holes were punched at the edge of the colony with a 5 mm-radius punch, and five clusters were placed in 100 mL of PDB and shaken at 180 rpm for 7 days at 28 °C before removal, and then filtered through eight layers of sterile gauze. This yielded an IAA concentration that was determined using Salkowski’s reagent [25], which was then compared with the IAA standard curve.

Transcriptomic analysis: Whole roots from treatments 1, 5, and 6 were sampled from pepper plants, snap-frozen in liquid nitrogen, and stored in a freezer at −80 °C. Total RNA was isolated and extracted from the tissues using Invitrogen TRIzol reagent, which was outsourced to Shanghai Personal Bio-technology Co., Ltd., Shanghai, China. Total RNAs of ≥1 μg were selected and enriched for poly (A) RNAs with Poly (A) RNA using Oligo (dT) magnetic beads, and cDNA synthesis was performed using RNA for cDNA synthesis. Purified size-selected and adapter-conjugated cDNA fragments were used for library construction. The mixed libraries were sequenced on the Illumina NovaSeq 6000 platform. Both raw and clean data generated from each library were sequenced to a depth of no less than 5 G. Raw reads obtained from the NovaSeq sequencing platform were filtered to remove shorter reads, junctions, and low-quality reads. All clean reads were localized to the pepper reference genome (http://peppergenome.snu.ac.kr/) using HISAT2 (v2.1.0) and statistically matched to each gene using HTSeq (v0.9.1). The differentially expressed genes (DEGs) were identified using the edgeR package, with DEGs defined as follows: |log2FoldChange|>1 and *p*-value <0.05. Kyoto Encyclopedia of Genes and Genomes (KEGG) enrichment analysis was performed using clusterProfiler (v4.6.0), and the threshold was set as a *p*-value <0.05.

### 2.4. Data Analysis and Visualization

Excel 2019 was used to handle the statistical data; SAS9.4 was used for one-way analysis of variance (ANOVA); and Duncan’s multiple comparison test was used to analyze the significance of differences, with *p* < 0.05 indicating a significant difference. Graphs were prepared using Origin 9.1 software.

## 3. Results

### 3.1. Antagonism and Identification of Biocontrol Microorganisms Against P. capsici

The antagonistic activity of the isolates against *P. capsici* was assessed using a double-culture test. Six biocontrol microorganisms from strains isolated from the inter-root soil of peppers demonstrated high antagonistic ability in the dual-culture test. The control strain of *P. capsici* grew uniformly in a radial direction on PDA (Figure 1a). In contrast, *P. capsici* mycelia were inhibited under the *Trichoderma* strain culture conditions, and colony expansion stopped, whereas *Trichoderma* strains continued to expand until they completely covered the *P. capsici* colonies and the whole plate (Figure 1b,c). All six biocontrol microorganism strains inhibited conidia of *P. capsici* by more than 70% (Figure 1d), with strain 6311 showing the strongest antagonistic effect on *P. capsici* filaments (reaching 82.22%). In addition, all six strains of biocontrol microorganisms covered the entire PDA plate with colonies of *P. capsici* after 5 days of incubation at 28 °C, and the production of *Trichoderma* strain conidia was observed in the medium.

Fungal isolates 2213 and 2221 produced smooth and spherical-to-subspherical conidia on the long main axis (Figure 2a,b and Appendix A). Meanwhile, 5111, 6311, and 6321 showed well-developed aerial mycelia on the PDA plate, and the colonies were fluffy to felted, white protruding, thick and fluffy, and formed clear concentric circles. After 5 days, the plates were covered with a large number of dark-green or yellowish-green spores, with a yellowish-brown backside (Figure 2c,d and Appendix A). Moreover, 7111 colonies were observed in the medium covering the plate. The fungal colonies of 7111 were white and light green, which turned dark green after 5 days (Appendix A). The ITS sequences of fungal isolates 2213, 2221, 5111, 6311, 6321, and 7111 were deposited in the GenBank database under the accession numbers SUB144343272213PP770682, SUB144343272211PP770683, SUB144343275111PP770684, SUB144343276311PP770685, SUB144343276321PP770686, and SUB144343277111PP770687. By using primers ITS4, ITS5, EF1-728F, and TEFl rev, the isolates 2213 and 2221 were molecularly identified as *T. harzianum*; 5111, 6311, and 6321 were identified as *T. brevicompactum*; and 7111 was identified as *T. virens*. These *Trichoderma* strains showed a greater than 90% homology with *T. harzianum* (MF871551.1), *T. brevicompactum* (KR094463.1), and *T. virens* (EU280080.1), respectively (Figure 2e,f, Appendix A and Appendix A).

### 3.2. Analysis of Disease-Suppressing and Growth-Promoting Properties of Biocontrol Microorganisms

#### 3.2.1. Analysis of Disease-Suppressing Properties of Biocontrol Microorganisms

The effect of the *Trichoderma* spp. on the biocontrol of *P. capsici* on isolated pepper fruits is shown in Figure 3. The uninoculated *P. capsici* treatment displayed no lesions, and no pathogenic mycelial growth was seen (Figure 3a), whereas the *P. capsici* treatment group inoculated with *P. capsici* alone exhibted visible lesions, with white mycelial growth. The diameters of the diseased spots on pepper fruits treated with the biocontrol microorganisms were all significantly smaller than those of the *P*. *capsici* treatment group (Figure 3d). Figure 3b,c indicate that the isolated strains of *Trichoderma* inhibited *P. capsici* growth in isolated pepper fruits. Among them, the 2213 and 6311 treatment groups displayed the smallest diameter of *P*. *capsici*-infested spots on the pepper fruits, with an average of 4.47 ± 0.03 cm and 5.23 ± 0.13 cm, and the inhibition rates were 40% and 29%, respectively (Figure 3n). The inhibition test of the *Trichoderma* fermentation solution showed that the fermentation solutions of six strains of *Trichoderma* exhibited different abilities to inhibit the mycelial growth of *P. capsici* (Figure 3e–g). Compared with the control, the fermentation solutions of strains 5111 and 6311 inhibited the mycelial growth of *P. capsici* by 100%. Meanwhile, the fermentation solutions of 6321 and 7111 also significantly reduced the growth of *P. capsici* mycelium, with inhibition rates of 69.72% and 27.52%, respectively. Meanwhile, the 2213 and 2221 fermentation broths achieved inhibition rates of only 4.59% and 8.26%, respectively (Figure 3o).

In the pot experiment, the *Trichoderma* strains also showed excellent pepper blight-suppressive effects (Figure 3h–m). Among the three treatments involving *P*. *capsici* inoculation, the disease indices of the 6311-added treatment groups were lower than those of the other two treatment groups (Figure 3p). Notably, the roots of peppers treated with 6311 displayed the smallest lesions (Figure 3m), and the control effect reached 55.56% (Figure 3q). This was followed by the appearance of watery spots on pepper roots inoculated with 2213 (Figure 3l). In the inoculation of blight alone treatment, the pepper stems showed greater shrinkage at the base (Figure 3k), and the leaves wilted and fell off, indicating entry into the necrotrophic stage.

#### 3.2.2. Analysis of the Growth-Promoting Properties of Biocontrol Microorganisms

The growth indexes of the effects of bioprophylactic microbial fermentation solution on pepper seeds are shown in Figure 4. Following the treatments of pepper seeds with six bioprophylactic microbial fermentation solutions, the seeds achieved a germination potential and a germination rate of 40.67–57.33% and 90.67–98.67%, respectively, which were significantly greater than those in the control group (Figure 4a). Among the six strains, 2213 and 6311 displayed the best growth-promoting effects, with a germination potential and a germination rate reaching 57.33% and 55.33% and 98.67% and 94.67%, respectively. In addition, the fermentation broths of the six biocontrol microorganisms exhibited significant germination-promoting effects on the embryonic roots and embryonic axes of pepper seeds (Figure 4c). The best germination-promoting effects were observed for the fermentation broths of 2213 and 6311, with the embryonic roots and embryonic axes in treatment 2213 increasing by 1.66-fold and 2.45-fold and those in 6311 by 1.50-fold and 2.21-fold compared with those of the control, respectively. The germination indices of the seeds treated with the six biocontrol microorganisms were significantly increased compared with those of the control treatment (Figure 4d), with 46.01% and 42.62% for 2213 and 6311, respectively. Meanwhile, the fresh weight of pepper seedlings increased by 47.62% and 42.86% compared with that of the control, respectively (Figure 4b). These results indicated that the *Trichoderma* fermentation solution improved the growth indexes of all pepper seedlings, with *T. harzianum* 2213 and *T. brevicompactum* 6311 exhibiting significant growth-promoting effects.

### 3.3. Mechanisms of Disease Suppression and Promotion by Biocontrol Microorganisms

#### 3.3.1. Physiological Evidence

Siderophores are capable of chelating iron to prevent the uptake of siderophores by plant pathogens [26], thus protecting plants by reducing the number of pathogens [27]. In this study, compared to unspliced fungi (Figure 5b), the *Trichoderma* 6311 treatment group showed a bright purple color on a double-layer chromogenic plate (Figure 5a), indicating the production of siderophores. The relative concentration of siderophores produced by different *Trichoderma* strains was determined (Figure 5c). The highest relative concentration of siderophores reached 72.88% for *T. brevicompactum* 6311, while 5111 and 6321 produced only small amounts of siderophores, and no siderophores were detected for *T. harzianum* 2213 and 2221. Light microscopy and SEM revealed the entangled parasitism of the biocontrol microbial mycelium on *P. capsici* hyphae and growth along the *P. capsici* mycelium, with a large number of branches entangled around the *P. capsici* filaments. In addition, the *P. capsici* idem were deformed or enlarged, indicating the infestation of its mycelium by 2213 (Figure 5f,g) and 6311 (Figure 5e,h–j). IAA, as an endogenous growth hormone that promotes plant growth and facilitates the formation of plant roots, plays an important role in plant growth and development [28]. Concentrations of IAA produced by biocontrol microorganisms are shown in Figure 5d. The findings showed that 2213, 2221, and 6311 produced high concentrations of IAA, namely, 30.91, 26.07, and 14.56 mg/L, respectively, whereas 5111, 6321, and 7111 produced only small amounts. High levels of IAA are known to promote lateral root formation [29] and increase root length [30].

#### 3.3.2. Transcriptomic Evidence

Among all the treatments, the proportion of Q20 and Q30 bases in the transcriptome of the pepper root was the lowest, with 98.87% and 96.69%, respectively (Appendix A). In the alignment results, at least 78.04% of the high-quality reads mapped successfully to the reference genome of CM334 (Appendix A). Principal component analysis of all transcripts showed that the first two principal components explained 82.9% of the total variation, and the three biological triplicates of each group clustered together (Figure 6a). Overall, 1346 pepper genes exhibited upregulated expression, and 829 genes displayed downregulated expression after treatment with strain 2213 (Figure 6b). A total of 930 pepper genes showed upregulated expression and 1363 genes displayed downregulated expression after treatment with strain 6311 (Figure 6b). Those treated with the 6311 strain exhibited 1107 upregulated genes and 2469 downregulated genes compared with the results for the treatment with strain 2213, indicating that the two types of *Trichoderma* had different effects on the pepper root system. To further understand the transcriptomic response of peppers to the two *Trichoderma* species, a Venn diagram of DEGs was produced. Overall, 7 genes (Figure 6c) and 16 genes (Figure 6d) were commonly upregulated and downregulated in the 2213 and 6311 strain treatments, suggesting that these DEGs are the key genes for the responses to the two *Trichoderma* species.

Among the 1346 upregulated genes in the strain 2213 treatment, there were 12 significantly enriched KEGG pathways, 11 of which were related to metabolism, including phenylpropane biosynthesis and zeatin biosynthesis (Figure 7a). Among the 930 upregulated genes in the strain 6311 treatment, there were 5 significantly enriched pathways, including phytohormone signal transduction and zeatin biosynthesis (Figure 7b). The seven co-upregulated genes in the strain 2213 and 6311 treatments are thought to be potentially involved in the defense response, suggesting that strains 2213 and 6311 could activate the defensive responses of peppers. Among the downregulated genes in the 2213 treatment group, phenylacetone biosynthesis, terpene skeleton biosynthesis, isoquinoline alkaloid biosynthesis, valine and leucine, and isoleucine degradation were particularly enriched. Meanwhile, phenylacetone biosynthesis, interconversion of pentose and glucuronic acid, and cyanoamino acid metabolism were enriched pathways associated with the 6311 downregulated genes (Figure 7c,d).

In both the strain 2213 and 6311 treatments, the growth hormone transporter protein (AUX22) genes (*CA06g16770* and *CA08g08660*) were upregulated (Appendix A); AUX22 genes are members of the Aux/IAA family of proteins. AUX22 is a short-lived protein localized in the nucleus, playing a crucial role in the IAA signal transduction pathway [31]. This suggests that treatments with strains 2213 and 6311 might promote pepper plant growth. In addition, two cellulose synthase (CesA) genes (*CA07g01680* and *CA02g13540*) and two xylem cysteine protease 1 (XCP1) genes (*CA12g18250* and *CA04g23350*) were uniquely upregulated in the strain 2213 treatment (Appendix A). CesA is usually associated with cell wall reinforcement to enhance structural resistance and improve disease defense [32]. Meanwhile, XCP1 is an important enzyme in plant plastid ectodomains and belongs to the class of xylose-like cysteine proteases [33]. It can activate defense responses by producing pathogen-associated molecular pattern (PAMP)-like peptides recognized by pattern-recognition receptors. The flavanone 3-hydroxylase (F3H) controlling genes (*CA08g16150* and *CA08g00130*) were upregulated in the stain 6311 treatments (Appendix A). Flavanone 3-hydroxylase (F3H) is a key enzyme that directs carbon flow to produce 3-hydroxylated flavonoids, including flavonols and anthocyanins. These substances can be used against pathogens such as those of *Colletotrichum sublineola* [34].

## 4. Discussion

*P. capsici* is one of the best known fungal pathogens and has serious adverse effects on crop yields globally, especially on peppers. Although chemical fungicides are available to counter this pathogen, they pose serious threats to human health and the environment [35]. With growing demands for sustainable agricultural development, biological control is considered to be an environmentally friendly and economical alternative to fungicides. *Trichoderma* strains are important representatives in the field of biological control, with significant practical value and potential for the biocontrol of plant diseases; thus, these have been studied worldwide. Indeed, the studies showed that *Trichoderma* strains can effectively inhibit approximately 30 plant pathogenic fungi, including *P. capsici*. In the present study, six strains of biocontrol *Trichoderma* with strong antagonistic activity against *P. capsici* were isolated from the rhizosphere soil of pepper plants collected from plots with long-term continuous cropping in Heibei Province, China. Morphological and molecular analyses led to the identification of two strains of *T. harzianum*, three strains of *T. brevicompactum*, and one strain of *T. virens*, with *T. brevicompactum* 6311 being identified as the strongest inhibitor of the mycelial growth of *P. capsici*. Fungal parasitism is a key means of biocontrol in species of *Trichoderma*, which can parasitize pathogenic microorganisms such as *Aspergillus rotundus* [18], *Mycosphaerella epidermidis* [15], and *Fusarium oxysporum* [36] and can invade or damage the mycelium, resulting in the enlargement, distortion, and rupture of the pathogen’s cells. Light microscopy and SEM of pathogenic fungal mycelium from the boundary of the zone of inhibition of dual-culture plates showed the mycelial parasitism of *T. brevicompactum* 6311 on *P. capsici*, suggesting that *T. brevicompactum* may attack the *P. capsici* cell wall/cell membranes and inhibit the growth of the *P. capsici* mycelium through the production of an iron-carrying antifungal substance, as well as through competition for space and nutrients [37,38].

It was previously reported that *Trichoderma* produces a variety of compounds with antagonistic properties, including a range of enzymes and metabolites such as cellulases, xylanases, pectinases, and chitinases [39]. Moreover, it has been shown that fermentation broths can be used to study the antagonistic capacity of *Trichoderma* metabolites. *Trichoderma* fermentation broth causes the malformation of pathogenic mycelium, with swollen, fragmented, and twisted hyphae. This study showed that the fermentation broth of isolate 6311 inhibited the growth of *P. capsici* mycelium by 100%, which was the first experimental result revealing that *T. brevicompactum* exhibited the highest inhibitory efficiency. Meanwhile, the fermentation broth of isolate 2213 showed little or no inhibitory effect. The reduction of the mycelial growth diameter in the isolated pepper fruit inhibition assay also demonstrated the ability of species of *Trichoderma* to inhibit the *P. capsici* mycelium, with isolates 2213 and 6311 achieving the most significant inhibition rates of 40% and 29%, respectively. Meanwhile, *Trichoderma* produces the growth hormone IAA [40], iron carriers [41], and other plant growth-promoting substances [42]. In this study, isolate 6311 was found to display the greatest ability to produce iron carriers, confirming its strong potential to inhibit pathogenic fungi. Meanwhile, for the first time, it was found that *T. brevicompactum* significantly enhanced the radicle-embryonic axis, fresh weight, germination rate, and germination index of pepper seedlings (Figure 4), which, to the best of our knowledge, has not been reported previously. Moreover, the results of the pot experiment showed that isolate 6311 could achieve a 55.56% prevention and control rate of pepper Phytophthora blight (Figure 3q). High-throughput sequencing technology was used for transcriptomic analysis on roots from treatment and control groups in which *Trichoderma* was applied alone. KEGG enrichment analysis of upregulated genes showed that genes related to the transcription factors AUX22 and AHP1, which were promoted by the application of isolate 6311, were generally upregulated. Growth hormones are central hormones for plant growth and development and are transported by a variety of transporter proteins with different biochemical and structural properties. Members of the Aux/IAA family have been identified as short-lived nuclear proteins that play a key role in suppressing the expression of ARF-activated genes [43,44]. The studies showed that the Aux/IAA genes exhibit high expression levels that may play roles in pepper fruit development and ripening, and AUX22 was identified as belonging to the Aux/IAA family. In addition, the AHP gene family is important for cytokinin signaling. Cytokinins are involved in the regulation of almost all important processes in plant growth and development. Cytokinin signaling is mediated through a mechanism involving continuous phosphorylation (phosphorelay). In *Arabidopsis*, receptors activated by cytokinins autophosphorylate and deliver the phosphate group to the phosphotransfer protein (AHP) and further to the downstream response regulator (ARR), thereby mediating cytokinin-regulated responses. In the current study, both AUX22 and AHP1 were upregulated in the isolate 2213 and 6311 treatment groups compared with the levels in the control group, indicating that both *Trichoderma* species have the ability to promote pepper growth. Meanwhile, genes related to XCP1 showed a separate upregulation in the treatment of isolate 2213. XCP1 is a key protease for plant immunity that activates systemic immunity by producing the cytokine CAPE9 from the classical salicylic acid signaling marker PR1. Genes related to XCP1 were individually upregulated in the isolate 2213 treatment [33]. Thus, *Trichoderma* 2213 has the ability to inhibit pepper Phytophthora blight. The F3H gene of the phenylpropane pathway was individually upregulated in the isolate 6311 treatment, and it has been shown that the phenylpropane biosynthetic pathway plays an important role in root resistance to pepper Phytophthora blight [45]. It was also found that the phenylpropane biosynthetic pathway leads to the synthesis of secondary metabolites involved in plant defense against pathogens.

## 5. Conclusions

In summary, *T. brevicompactum* 6311 was isolated from the rhizosphere soil of peppers in this study. Experimentally, it was proven that in terms of disease suppression, isolate 6311 could effectively inhibit the growth of *P. capsici* mycelia on both plates and in pepper fruit, and the metabolites produced in its fermentation broth could also inhibit *P. capsici*. Moreover, in potting experiments, isolate 6311 demonstrated excellent preventive and control effects on pepper Phytophthora blight. In terms of growth promotion, the sterile fermentation broth of isolate 6311 promoted the growth of pepper seedlings. By exploring the mechanisms involved, it was found that isolate 6311 inhibited the growth of pepper Phytophthora blight mycelia and improved its controlling effect by secreting antimicrobial substances, performing parasitism, and producing iron carriers. It was also revealed that its ability to secrete IAA could effectively improve the growth indexes of pepper seedlings. In addition, the results of transcriptomic sequencing analysis on the pepper root system showed that isolate 6311 affected the gene expression in the pepper root system and regulated the promotion of pepper growth, resistance-related signal transduction, and phytohormone pathways, such as the growth hormone pathway and the phenylpropane biosynthesis pathway. Our study provides a potential biocontrol agent for the biocontrol of *P. capsici* in peppers and offers possible insights into the mechanism of interaction between biocontrol agents and pathogens. In future work, we plan to focus on the application of isolate 6311 for biocontrol purposes in agricultural production.

## Figures and Tables

**Figure 1 jof-11-00105-f001:**
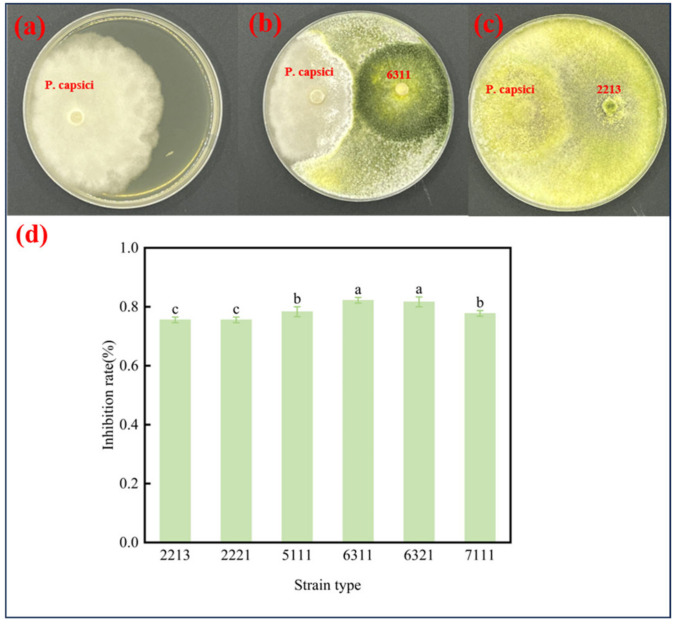
Biocontrol microorganisms and *P. capsici* dual-culture test and developmental tree of strains after 7 days of inoculation. (**a**) Inoculation with *P. capsici* only; (**b**) inoculation with *Trichoderma* 6311 and *P. capsici*; (**c**) inoculation with *Trichoderma* 2213 and *P. capsici*; (**d**) inhibition of *P. capsici* by the biocontrol microorganisms. Identical letters denote no significant differences between the respective groups (*p* > 0.05), whereas differing letters indicate a significant difference (*p* < 0.05).

**Figure 2 jof-11-00105-f002:**
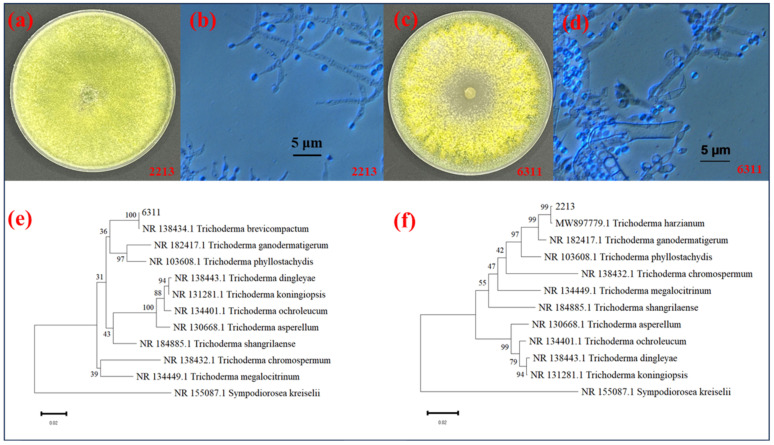
Morphological and molecular identification of biocontrol microorganisms. (**a**,**b**) *Trichoderma* 2213 colony morphology and spore morphology; (**c**,**d**) *Trichoderma* 6311 colony morphology and spore morphology; (**e**,**f**) 2213 and 6311 developmental trees.

**Figure 3 jof-11-00105-f003:**
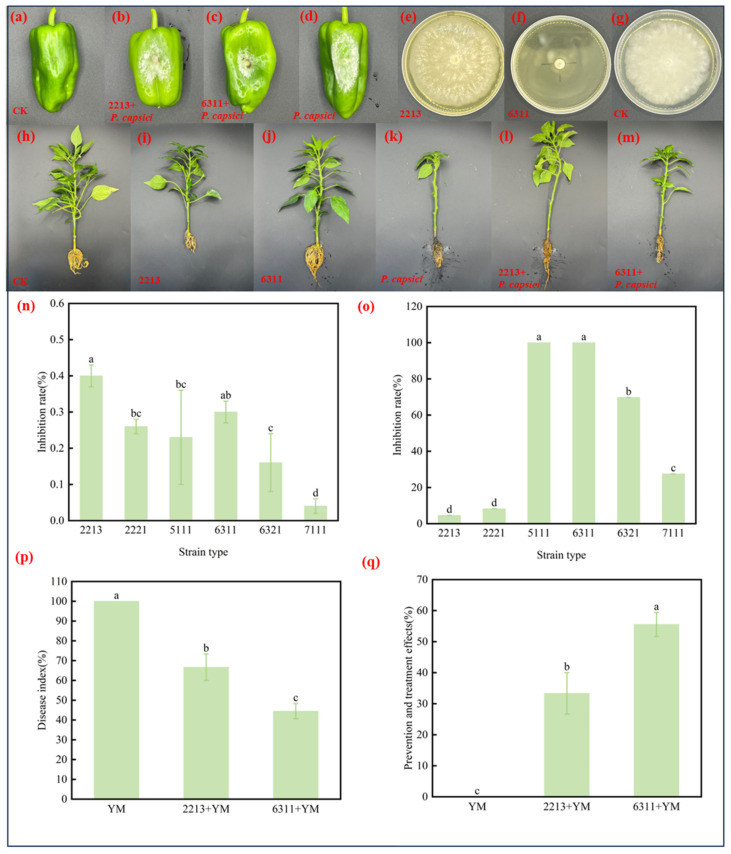
Results of pepper fruit or plant potting experiment. (**a**) CK group without any added microorganisms; (**b**) 2213+ *P. capsici* treatment group; (**c**) 6311+ *P. capsici* treatment group; (**d**) *P. capsici* treatment group; (**e**) no fermentation solution inhibition of *P. capsici*; (**f**) 2213 fermentation solution inhibition of *P. capsici*; (**g**) 6311 fermentation solution inhibition of *P. capsici*; (H) normal growth of *P. capsici* without fermentation solution; (**h**–**m**) growth of pepper after 5 days of each treatment; (**h**) CK; (**i**) 2213; (**j**) 6311; (**k**) *P. capsici*; (**l**) 2213+ *P. capsici*; (**m**) 6311+ *P. capsici*; (**n**) inhibition of *P. capsici* in pepper fruits; (**o**) *Trichoderma* fermentation broth inhibition; (**p**) disease indices for each treatment in the potting experiment; and (**q**) preventive and control effects of each treatment. Identical letters denote no significant differences between the respective groups (*p* > 0.05), whereas differing letters indicate a significant difference (*p* < 0.05).

**Figure 4 jof-11-00105-f004:**
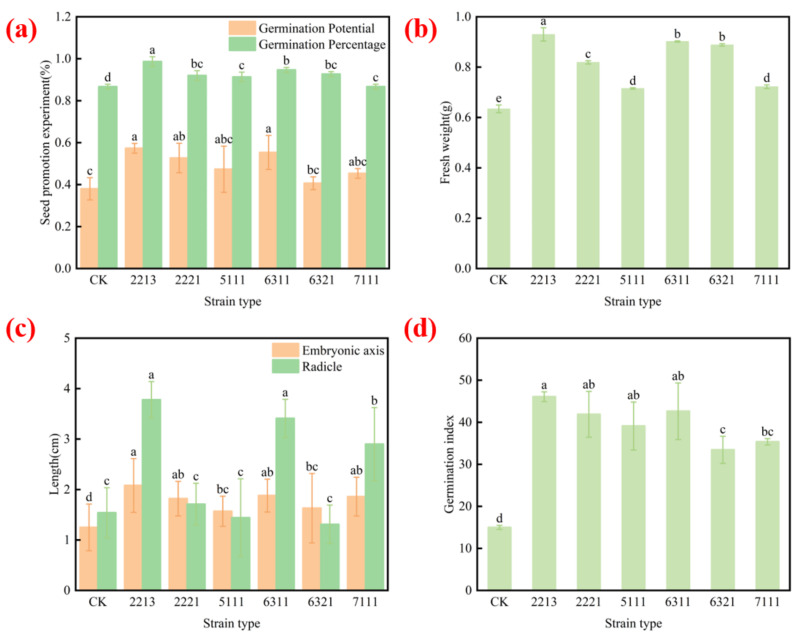
Effect of biocontrol microbial fermentation solution on the growth of pepper seedlings, where CK stands for no fermentation solution applied. (**a**) Seed germination potential and germination rate; (**b**) fresh weight; (**c**) lengths of embryonic axis and radicle; (**d**) germination index. Identical letters denote no significant differences between the respective groups (*p* > 0.05), whereas differing letters indicate a significant difference (*p* < 0.05).

**Figure 5 jof-11-00105-f005:**
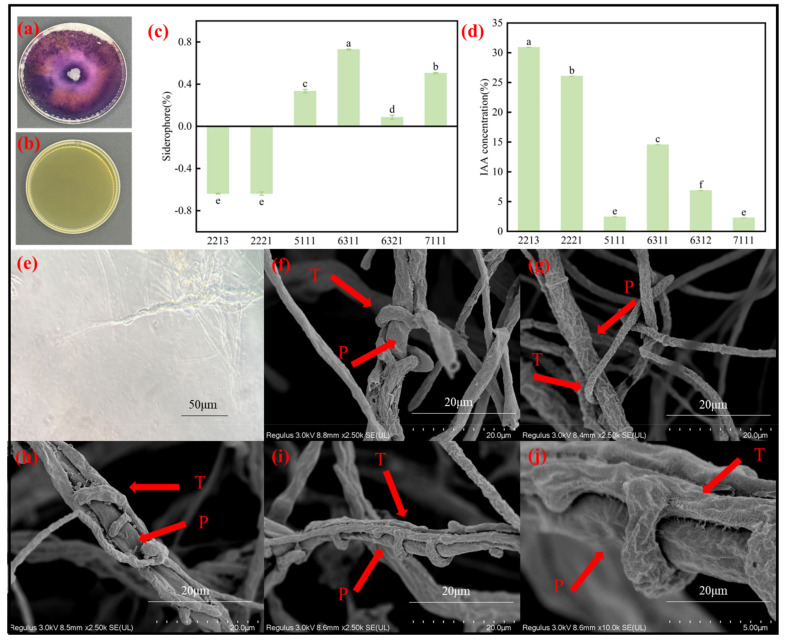
(**a**,**b**) Bilayer plates inoculated with 6311 and not inoculated with fungi; (**c**,**d**) relative concentration of siderophores and concentration of IAA upon *Trichoderma* treatment, respectively; (**e**) 6311 (T) hyphae entangled in *P. capsici* (P) hyphae under a 400× optical microscope; (**f**,**g**) SEM images of 2213 (T) hyphae entangled in *P. capsica* (P) hyphae; (**h**–**j**) SEM images of 6311 (T) hyphae entangled in *P. capsica* (P) hyphae. Identical letters denote no significant differences between the respective groups (*p* > 0.05), whereas differing letters indicate a significant difference (*p* < 0.05).

**Figure 6 jof-11-00105-f006:**
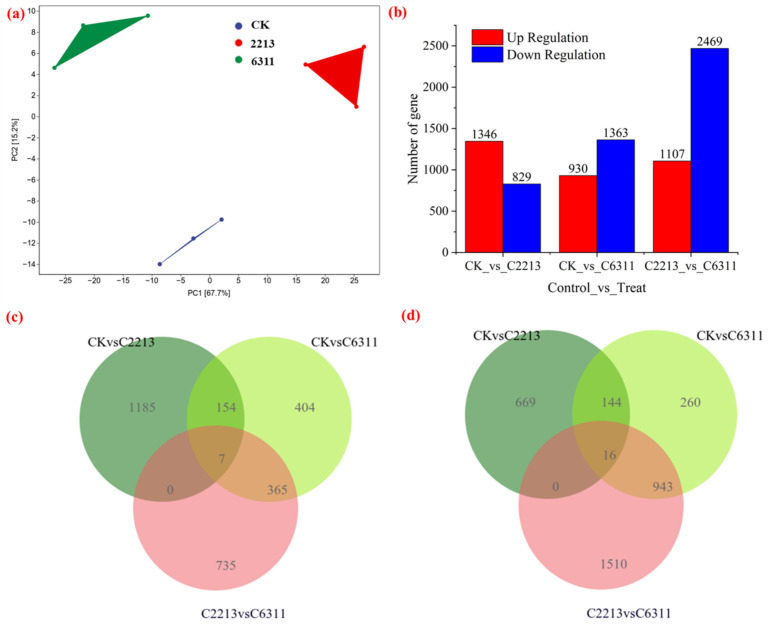
Sample correlations, and Venn diagrams of differentially expressed genes (DEGs). (**a**) The PCA for all samples; (**b**) the DEGs of all samples; (**c**,**d**) Venn diagrams of upregulated and downregulated DEGs, respectively.

**Figure 7 jof-11-00105-f007:**
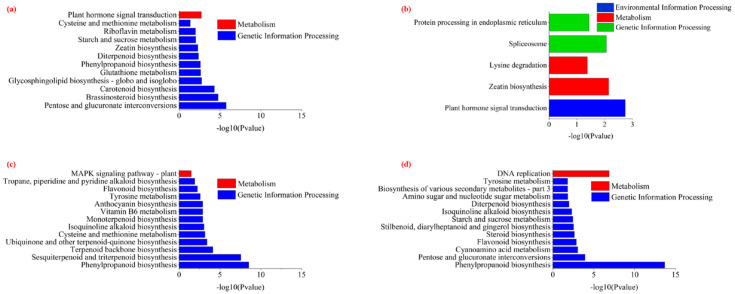
KEGG enrichment analysis of DEGs. (**a**,**b**) KEGG enrichment analyses of upregulated and downregulated genes in strain 2213 treatment, respectively; (**c**,**d**) KEGG enrichment analysis of upregulated and downregulated genes in strain 6311 treatment, respectively. Identical letters denote no significant differences between the respective groups (*p* > 0.05), whereas differing letters indicate a significant difference (*p* < 0.05).

## Data Availability

The original contributions presented in this study are included in the article/Appendix A. Further inquiries can be directed to the corresponding author(s).

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
