# Peer review of "Trichoderma brevicompactum 6311: Prevention and Control of Phytophthora capsici and Its Growth-Promoting Effect"

_jof, 2025, doi:10.3390/jof11020105_

Round 1

Reviewer 1 Report

The manuscript presents the results of isolating and screening six biocontrol strains of Trichoderma with high antagonistic activity against P. capsici from the rhizosphere soil of healthy peppers. The manuscript provides valuable information on the biological control of P. capsici. The authors implemented appropriate methodologies for biocontrol isolation, detecting and proving their biocontrol abilities.

However, the manuscript has weaknesses in the detection of Trichoderma species. They used only ITS region primers. For accurate species diagnosis, they must apply a second primer pair that amplifies different genes in Trichoderma and perform blastn analysis.

The other important comments are listed here. The minor corrections are in the pdf text and attached to the report.

1.     In the abstract all Trichoderma species must be included with their numbers.

2.     In disease names like “pepper phytophthora blight”, it is not necessary to start with a capital and make it italic the phytophthora name. This rule has to be applied throughout the text.

3.     Figure legends particularly Fig 2 have to be corrected. Figure legends have to be self-explanatory. Because of that the abbreviations have to be described by writing full.

4.     Figure 2 a, b, c, d…. descriptions in the text are wrong.

5.     The scientific names have to be written at first mention and afterward the genus name is abbreviated. This rule has to be applied throughout the text.

6.     In section 3.2.1 of the manuscript, there are several unclear sentences, which are detailed in the attached file. These sentences need clarification.

7.     In the discussion section “…. but they pose a serious threat to human health and the environment. …” . It could be mentioned here resistance evaluation of the phytophthora to pesticides.

8.     In the text there is an emphasis on “short dense mycorrhizal fungus”. This emphasis has to be clarified in the manuscript.

The manuscript presents the results of isolating and screening six biocontrol strains of Trichoderma with high antagonistic activity against P. capsici from the rhizosphere soil of healthy peppers. The manuscript provides valuable information on the biological control of P. capsici. The authors implemented appropriate methodologies for biocontrol isolation, detecting and proving their biocontrol abilities.

However, the manuscript has weaknesses in the detection of Trichoderma species. They used only ITS region primers. For accurate species diagnosis, they must apply a second primer pair that amplifies different genes in Trichoderma and perform blastn analysis.

The other important comments are listed here. The minor corrections are in the pdf text and attached to the report.

1.     In the abstract all Trichoderma species must be included with their numbers.

2.     In disease names like “pepper phytophthora blight”, it is not necessary to start with a capital and make it italic the phytophthora name. This rule has to be applied throughout the text.

3.     Figure legends particularly Fig 2 have to be corrected. Figure legends have to be self-explanatory. Because of that the abbreviations have to be described by writing full.

4.     Figure 2 a, b, c, d…. descriptions in the text are wrong.

5.     The scientific names have to be written at first mention and afterward the genus name is abbreviated. This rule has to be applied throughout the text.

6.     In section 3.2.1 of the manuscript, there are several unclear sentences, which are detailed in the attached file. These sentences need clarification.

7.     In the discussion section “…. but they pose a serious threat to human health and the environment. …” . It could be mentioned here resistance evaluation of the phytophthora to pesticides.

8.     In the text there is an emphasis on “short dense mycorrhizal fungus”. This emphasis has to be clarified in the manuscript.

Author Response

Thank you for your suggestion. I have made minor corrections to the PDF text. Below is my response to your comments.

Comments 1.In the abstract all Trichoderma species must be included with their numbers.。

Reply 1: Thank you for your suggestion. The full name of T. brevicompactum 6311 has been added to the abstract and checked and verified throughout the entire text.

Comments 2.In disease names like “pepper Phytophthora blight”, it is not necessary to start with a capital and make it italic the Phytophthora name. This rule has to be applied throughout the text.

Reply 2: Thank you for your suggestion. I have consulted other related articles and found that Phytophora needs to be capitalized. Therefore, we have checked the entire text and modified "Pepper Phytophora Blight" to "pepper Phytophora blight".

Comments 3.Figure legends particularly Fig 2 have to be corrected. Figure legends have to be self-explanatory. Because of that the abbreviations have to be described by writing full.

Reply 3: Thank you for your suggestion. We have revised the legend of Figure 1 and Figure 2 to lines 283-284, 304, and 335-342, providing a detailed explanation of the time for determining the results of the dual culture experiment, the control processing information in the fermentation broth antibacterial test, and some detailed information.

Comments 4.Figure 2 a, b, c, d…. descriptions in the text are wrong.

Reply 4: Thank you for your suggestion. The text in figures 2a, 2b, 2c, 2d, etc. has been modified.

Comments 5.The scientific names have to be written at first mention and afterward the genus name is abbreviated. This rule has to be applied throughout the text.

Reply 5: Thank you for your suggestion. We have conducted a full-text check and have abbreviated the second occurrence of the genus name.

Comments 6.In section 3.2.1 of the manuscript, there are several unclear sentences, which are detailed in the attached file. These sentences need clarification.

Reply 6: Thank you for your reminder. I have revised the part you mentioned, please refer to lines 327-331 in the text for details

Comments 7. In the discussion section “…. but they pose a serious threat to human health and the environment. …” . It could be mentioned here resistance evaluation of the Phytophthora to pesticides.

Reply 7: Thank you for your suggestion. I have added the evaluation of plant pathogen resistance to pesticides in lines 451-453 of the original text and discussed it in conjunction with relevant literature.

Comments 8.In the discussion section “…. but they pose a serious threat to human health and the environment. …” . It could be mentioned here resistance evaluation of the Phytophthora to pesticides.

Reply 8: Thank you for your suggestion. This is a translation error and has been corrected to T. brevicompactum.

Reviewer 2 Report

Dear authors

 I have reviewed the manuscript ID jof-3369661 entitled: “Efficient management of biological contaminants from Phytophthora: Role and effects of the neglected Trichoderma brevicompactum”.

I consider that the work is very interesting because it addresses a very current subject as the search alternatives ecofriendly to reduce the use of agrochemicals to plant disease management. It is important to highlight that the use of species de Trichoderma are widely known as biological control agents and also to promote plant growth.

I consider that the research carried out by the authors provides a potential alternative for the management of Phytophthora blight on pepper caused by Phytophthora capsici.

Dear authors

First of all, I apologize if my English is not correct because my mother tongue is Spanish.

Among my comments, I first refer to the manuscript title because I don't find it very scientific or representative of the research. It is not appropriate and is too broad.

Efficient management of biological contaminants from Phytophthora: Role and effects of the neglected Trichoderma brevicompactum”.

 I also believe that it should be clarified that the isolates obtained from the rhizosphere soil of peppers are potential biocontrol agents, not biocontrol agents that were isolated. The methodology should also be improved. Initially, authors not mention the number of isolates obtained from the soil and how many isolates were selected and based on what criteria to test as biocontrol agents.

In many experiments the treatments used are not clearly stated. An also, for example, it is not mentioned how the conidial suspensions were adjusted for inoculation….. 

Furthermore, some other comments are as follows:

According to mentioned above I consider that the manuscript could be accept to publish in Journal of Fungi, after modifications.

 I am listed below other suggestions and comments that I think should be made in the text to improve or clarify the manuscript for publication.

 Abstract

I suggest to mention the strain of Thichoderma as strain 6311 or otherwise and not just 6311.

… Scanning electron microscopy revealed that 6311 achieved reparasitism of P. capsici, producing siderophores and the growth hormone indoleacetic acid (IAA) to achieve disease-suppressive and growth-promoting functions…

Reparasitism???

How can the production of indolacetic acid be demonstrated through the scanning electron microscope?

The molecular identification of the strain of Trichoderma as T. brevicompactum is not mentioned in the abstract.

Keywords: “soil health”: I do not consider that this keyword is representative of this work.

Introduction

I think they could initially clarify that in vitro and in vivo tests were carried out, mentioning them separately.

Materials and Methods

I suggest to include subtitles, for example: Source of biocontrol fungi and pathogenic strains and

Isolation and screening of biocontrol fungi.

Two batches of inter-root soil of healthy pepper plants were collected from long-term continuous crop-ping plots during the blight outbreaks and stored in a 4°C incubator to cultivate biocontrol fungi and isolate and purify biocontrol fungi P. capsici was provided by the Institute of Plant Protection, Hebei Academy of Agricultural and Forestry Sciences, Hebei Province, China.

I suggest shortening the sentence.

-Isolation and screening of biocontrol fungi.

Five grams of fresh inter-root soil was placed in 45 mL of sterile water and shaken on a shaker at 28°C and 180 rpm for 1 h. The soil samples were diluted to 10−1, 10−2, 10−3, 10−4, 10−5, and 10−6 using the stepwise dilution method, and 100 μL of each soil suspension was aspirated on potato dextrose agar (PDA) medium plates for coating;….

The suspension was aspirated on PDA medium for coating : I suggest to check this sentence.

-Bioculture test of biocontrol fungi and P. capsici.

A 5-mm-radius sterilized hole punch was used to punch holes at the edge of the P. capsici colonies cultured for 5 days. The P. capsici cake was inoculated on one side of a 9-cm-diameter PDA plate, and the symmetrical position of the P. capsici block was inoculated with the biomicrobe block to make the distance between the blocks 5 cm.

Cake and block are not usually used, normally we used disks or sections of the edge of the colonies developed on PDA …

The plate was then sealed with sealing film and subjected to inverted incubation for 7 days in the dark at 28°C.

Wasn't the test done in Petri dishes?

How the authors guarantee sterility?

 -        Pag. 3.  2.2. Biocontrol microorganisms inhibit disease and promote growth performance test

In vitro inhibitory activity test of fermentation solution against P. capsica. Activated Trichoderma PDA plates were punched into conical flasks of potato dextrose liquid medium (PDB), and cultured by oscillation at 180 rpm and 28°C for 7 days.

The authors use terminology that is not common in phytopathology, for example activated Trichoderma PDA plates. The sentence could be “colonies of strains of Trichoderma developed on PDA were inoculated into conical…..

How many isolated strains of antagonistic fungi were used?

-Fermentation solution on pepper promotion test.

Clarify treatments.

-Pepper fruit blight antagonism test

…A sterile perforator was used to create a wound of 5 mm in diameter and 1 mm in depth at the waist of the pepper biocontrol microbial cakes were inoculated into the wounds, the wounds were covered with blotting paper that absorbed a sufficient amount of sterile water, and the fruits were placed in sterilized plastic pots covered with film to keep them warm and moisturized. After 24 h of incubation at room temperature, the fungal cake was removed, and then P. capsici was inoculated into the wounds to continue incubation. The diseased area was observed and photographed every day, and this area was determined, with each treatment being repeated three times.

Clarify treatments.

How the pathogen was inoculated?

- Experimental method: Three peppers of TianYu No. 5 were planted in each pot, cultivated for 90 days followed by wounding of the roots, and, in treatments 2, 3, 5, and 6, 10 mL of Trichoderma spore solution with a spore count of 1 x106 was added to the pepper’ root systems.

How many strains of Trichoderma were tested?

How did they make the wounds on the roots?

Spore solution change for conidial suspension..

A hematocimeter was used to count spores, how was it done?

After 5 days of Trichoderma colonization, 10 mL of Phytophthora capsici spp. spore solution with a spore count of 1 105 was added in treatments 2, 3, and 4, with each treatment being replicated three times.

spp. Remove.

I suggest for example: Five days after inoculation with Trichoderma suspensions, a suspension of 1x 105 conidia of P. capsici was added in…

Blight disease condition index = 𝜮Disease level of diseased plants × number of diseased plants

                                                             Highest incidence level x total number of plants surveyed

Authors evaluated disease severity not incidence. They are two different parameters for disease estimation in phytopatometry.

-2.3. Analysis of disease-suppressing and growth-promoting mechanisms of biocontrol microorganisms

What were the Trichoderma strains used?

-Siderophores performance test. The biocontrol microbial strains were cultured on PDA plates for 7 days to grow into colonies. Then, the cooled CAS test medium was poured onto the PDA plates and left to stand for 24 h to observe the plate color. Each treatment was repeated three times. What were the treatments?

-IAA growth-promoting properties. The bioprophylactic microorganisms were activated on PDA plates at 28°C for 5 days. Holes were punched at the edge of the colony with a 5-mm-radius punch, and five clusters were placed in 100 mL of PDB and shaken at 180 rpm for 7 days at 28°C before being removed, and then filtered through eight layers of sterile gauze.

I suggest to check this sentence with references to describe the methodology.

-Identification of biocontrol microbial strains. (1) The strains to be identified were inoculated onto PDA plates and cultured at a constant temperature of 28°C. After 5 days, the morphology and color of the colonies were observed, and the morphological characteristics of conidia and conidial peduncles were observed under a light microscope.

Morphological characteristics of Trichoderma spp. are conidia, phialides and conidiophores. Peduncles it is not a used term.

- Molecular biological identification. I suggest to eliminate “biological”. Also, with respect to the identification, it is widely known that the identification of Trichoderma species is very difficult and that more of ITS regions are required to determine at level species.

-3.1. Antagonism and identification of biocontrol microorganisms against P. capsici

The antagonistic activity of the isolates against P. capsici was assessed using a double-culture test. Six biocontrol microorganisms among the isolated strains demonstrated high antagonistic ability in the dual-culture test.

Six isolated strains were used in the assays? this is not mentioned anywhere in Materials and Methods.

…In contrast, P. capsici filaments were inhibited under Trichoderma spp. culture conditions, and colony expansion stopped, whereas Trichoderma spp. continued to expand until it completely covered the P. capsici colonies and the whole plate (Fig. 1b, 1c).

The world correct is mycelia not filaments…

…All six biocontrol microorganism strains inhibited P. capsici by more than 70% (Fig. 1d), with strain 6311 showing the strongest antagonistic effect on P. capsici filaments (reaching 82.22%).

Microorganism strain is not correct. I suggest changing spore to conidia.

-Moreover, 7111 was observed to be covered by Trichoderma colonies in the medium.

This sentence is not correct. 7111 is a strain of Trichoderma. Please check this sentence.

 -Pag. 7. In the pot experiment, the Trichoderma strains also showed excellent pepper blight-suppressive effects, and the pepper plants inoculated with Trichoderma 2213 and 6311 and not inoculated with Trichoderma exhibited no disease symptoms….

"Clarify which ones were not inoculated with Trichoderma."

Results

Figure 4. Change filaments by hyphae

Discussion

Pag. 12.

In the present study, six strains of biocontrol Trichoderma with strong antagonistic activity against P. capsici were isolated from the rhizosphere soil of pepper plants collected from plots with long-term continuous cropping in Heibei Province, China.

Six strains were isolated which demonstrated strong antagonistic activity against P. capsici.

Others comments have been made in the manuscript.

Author Response

Comments 1. The author should clarify that the isolate obtained from the rhizosphere soil of chili peppers is a potential biological control agent, rather than an isolated biological control agent. The quantity of separated substances obtained from the soil should be supplemented, the number of separated substances selected, and the criteria used as biological control agents.

Reply 1: Thank you for your suggestion. We have preliminarily isolated 6 strains with Phytophthora capsici biocontrol function through double culture test, but the expression in the text does exist the situation of mixed use of biological control agents and potential biological control agents. Therefore, the text is now uniformly revised as follows: the 6 strains with Phytophthora capsici biocontrol function have been preliminarily isolated as potential biological control agents. After the comparative analysis of the two functions of disease resistance and growth promotion of the 6 strains, specifically through fermentation broth inhibition test, in vitro fruit disease resistance test, seed germination test, pot test, etc., the biocontrol bacteria of T. brevicompactum 6311 with dual functions of disease inhibition and growth promotion have been finally determined as our final biological control agent. Relevant contents have been modified and supplemented in many places, such as lines 102-105.

Comments 2. I suggest using the full name of strain 6311. How to prove the production of indole-3-acetic acid through scanning electron microscopy? The abstract does not mention that the molecular identification of the Trichoderma strain is T brevicompactum.

Reply 2: Thank you for your suggestion. I have changed 6311 to T. brevicompactum 6311, which has been unified in the full text. Scanning electron microscope pictures are used to speculate the possible mechanism of disease inhibition, and the production of indoleacetic acid is determined by adding salkowski reagent after the strain is cultured in PDB medium. The process and method of determination are supplemented in this paper. See lines 236-237 for details. In addition, the identification results of morphology and molecular biology of T. brevicompactum 6311 were added in the abstract.

Comments 3. The keyword 'soil health' cannot represent this work.

Reply 3: Thank you for your suggestion, it has been deleted.

Comments 4. I think we can first explain that in vitro and in vivo experiments were conducted

Reply 4: Thank you for your suggestion. I have made the changes in lines 86-92.

Comments 5. I suggest adding a subtitle.

Reply 5: Thank you for your suggestion. According to the opinions of two reviewers, the subtitle and expression have been comprehensively modified.

Comments 6. I suggest briefly introducing the sources of biological control fungi and pathogenic strains

Reply 6: Thank you for your suggestion. Based on the comments of two reviewers, the whole sentence has been revised. First, adjust to short sentence expression. Second, sort out the sentence logic. See line 102-105 for details.

Comments 7. Suggest checking this sentence: Suck the suspension onto PDA medium for coating.

Reply 7: Thank you for your suggestion. We have changed 'for coating' to 'for spreading'.

Comments 8. What is the process of double culture test and how does the author ensure that the process is sterile?

Reply 8: Thank you for your suggestion. The process of double culture test is as follows: select the evenly developed colony edge of Phytophthora capsici or Trichoderma strains on PDA medium, use sterilized puncher to punch holes on the colonies, carefully transfer 1 piece of Phytophthora capsici and 1 piece of Trichoderma strains to PDA plate, and then seal the plate with sealing film at a symmetrical position 5cm away, and culture upside down in the dark at 28 ° C. Aseptic operation was strictly carried out during the test. All materials used in the test, such as puncher, culture medium and sealing film, were sterilized with alcohol lamp. All operations were carried out in the ultra clean workbench. It is supplemented and explained in the corresponding part of the article. See lines 116-122 for details.

Comments 9. The terminology used by the author is not common in plant pathology, such as activated Trichoderma strains PDA plates. This sentence can be "Inoculate the colony of Trichoderma strains cultured on PDA into a cone...". How many isolated antagonistic fungi were used?

Reply 9: Thank you for your suggestion. The sentence has been modified in line 144-146 according to your suggestion, and 6 strains of antagonistic fungi have been added.

Comments 10. Clarify the treatment method in the experiment of promoting the effect of fermentation broth on pepper.

Reply 10: Thank you for your suggestion. The specific treatment method is as follows: disinfect the healthy and consistent pepper seeds with 5% NaClO for 5 minutes, rinse them fully with sterile water, drain the water, and then put them into the 5-fold diluted fermentation broth of T. brevicompactum 6311 for 24 hours. Put the seeds evenly in the germination box paved with double-layer sterile germination paper, place 50 seeds in each box, and culture them in the moist incubator at 28 ° C for 10 days. The operation of the other five potential biocontrol bacteria is the same, and repeat each for 3 times. Relevant sentences have been modified on line 153-156.

Comments 11. Clarify the processing methods for chili fruit experiments and how pathogens are inoculated?

Reply 11: Thank you for your suggestion. In the in vitro fruit test, the specific process of pathogen inoculation is as follows: cultivate Phytophthora capsici on PDA plate for 5 days, punch a 5mm diameter sterile punch at the edge of the colony with uniform growth to obtain a complete bacterial block, and then inoculate the bacterial block on the wound of pepper fruit for infection. The corresponding supplements and references are cited in this paper. See lines 170-172 for relevant contents.

Comments 12. How many types of Trichoderma strains were tested in the potted plant experiment? How do they cause wounds at the roots? Changes in spore solution of conidia suspension? How is spore counting done using a hemocytometer? I suggest modifying the sentence in the pot experiment to add a spore suspension with a spore count of 1 × 105 in treatments 2, 3, and 4 after inoculation with Trichoderma strains suspension for 5 days.

Reply 12: Thank you for your suggestion. Six treatments were set in the pot experiment: blank control, T. brevicompactum 6311, T. brevicompactum 6311+Phytophthora, Phytophthora, T. harzianum  2213, T. harzianum 2213+Phytophthora. Therefore, two kinds of Trichoderma strains were tested in the pot experiment. The operation of root injury test is as follows: use the sterilization blade to make a wound about 1mm deep at the root 1cm below the stem base of all pepper plants, and ensure that the position, depth and time of root injury of all plants are consistent. The spore concentrations of conidia suspensions of T. brevicompactum 6311 and T. harzianum 2213 ranged from 1×108 to 5×108, respectively. However, we diluted the spore suspensions during inoculation. The specific process of counting the spore numbers of the two Trichoderma strains species was as follows: preliminarily estimate the approximate concentration of spore stock solution under the microscope, dilute the high concentration spore solution to the corresponding multiple with sterile water, and add 300 μL of dilution solution to the blood cell counting plate containing the cover glass. After the spore solution migrated and balanced in the counting plate, select 5 of the 25 cells under the microscope, and accurately count the spore solution concentration. The corresponding content and suggested perfect sentences are also supplemented appropriately in the text. See lines 188-190 for details.

Comments 13. The authors assessed the severity of the disease, not the incidence rate. These two parameters are two different parameters for estimating diseases in plant morphometrics.

Reply 13: Thank you for your suggestion. I have replaced the incidence with severity in the calculation formula in the pot experiment, which is 200 lines.

Comments 14. What strains of Trichoderma were used in experiments on the disease inhibition and growth promotion mechanisms of microorganisms in biological control?

Reply 14: Thank you for your suggestion. The comparative analysis of 6 Trichoderma strains was carried out in the double culture test, in vitro fruit test, fermentation broth antibacterial test, fermentation broth growth promoting test, iron carrier and IAA detection. Two kinds of Trichoderma tests, T. brevicompactum 6311 and T. harzianum 2213, were carried out in the pot experiment.

Comments 15. What are the processing groups in the performance testing of siderophores?

Reply 15: Thank you for your suggestion. In this experiment, there are six treatments and six potential biocontrol strains: T. harzianum 2213, T. harzianum 2221, T. brevicompactum 5111, T. brevicompactum 6311, T. brevicompactum 6321 and T. virens 7111. Each treatment is repeated three times and has been supplemented appropriately in rows 221-225.

Comments 16. I suggest adding references to the characteristics of IAA promoting growth to describe the method.

Reply 16: Thank you for your suggestion. I have cited a relevant literature in lines 236-237 and introduced the differences in experimental methods compared to the literature.

Comments 17. The morphological characteristics of the genus Trichoderma are conidia, phialides and conidiophores.. Peduncles it is not a used term.

Reply 17: Thank you for your suggestion. I have checked the entire paragraph and made revisions at line 131.

Comments 18.Molecular biological identification. I suggest to eliminate “biological”. Also, with respect to the identification, it is widely known that the identification of Trichoderma species is very difficult and that more of ITS regions are required to determine at level species.

Reply 18: Thank you for your advice. According to experts' suggestions, we adopted the expression of molecular identification and modified it in the corresponding position. Based on the existing literature on the identification of Trichoderma strains species, we found that many studies used its gene as a primer for the identification of Trichoderma strains species, such as: T. reesei and Aspergillus awamori (naher, L.; fatin, S.N.; sheikh, M.A.H., et al. cellular enzyme production from filimentous fundi T. reesei and Aspergillus awamori in merged Federation with rice straw. Journal of fundi, 2021, 7 (10): 868.), T. harzianum aym3 (madbouly, A.K.; Rashad, y.m.); Ibrahim, M.I.M., et al. biodegradation of aflatoxin B1 in maze grains and suppression of its biosynthesis related genes using endophytic Trichoderma harzianum aym3. Journal of fundi, 2023, 9 (2): 209.), T. asperellum (khuong, n.q.; nhien, D.B.; Thu, l.t.m., et al. using Trichoderma asperellum to antonioze lasiodiplod IA theobromae causing stem end rot disease on pomelo (Citrus maxima). Journal of fungi, 2023, 9 (10): 981). The common point of these studies is that only its gene is used to identify fungal species. The journals published by these studies represent a professional and high level. Of course, we are also willing to use more specific primers to supplement the identification of Trichoderma strains, but it requires more test time to ensure the identification work.

Comments 19. In the dual culture experiment, it was not mentioned that six biological control microorganisms in the isolated strains showed high antagonistic ability.

Reply 19: Thank you for your suggestion. What I want to express is that we have obtained six microorganisms with biological control potential through dual culture experiments, which have been modified at lines 116-122.

Comments 20. I suggest changing the spore in the sentence '6311 strain has the strongest antagonistic effect on Phytophthora capsici filaments (reaching 82.22%)' to 'mycelia '.

Reply 20: Thank you for your suggestion. I have made the changes on line 272-276.

Comments 21. 7111 was observed to be covered by Trichoderma colonies in the culture medium. This sentence is incorrect. 7111 is a strain of Trichoderma. Please check this sentence.

Reply 21: Thank you for your suggestion. I have revised the sentence as moreover in lines 291-292, 7111 colonies were observed in the medium covering the plate.

Comments 22. Clarify which ones were not inoculated with Trichoderma in the potted plant experiment

Reply 22: Thank you for your suggestion. Six treatments were set in the pot experiment: treatment 1, blank control (without any bacteria); Treatment 2, T. harzianum 2213+Phytophthora infestans; Treatment 3, T. brevicompactum 6311+Phytophthora infestans; Treatment 4, Phytophthora infestans; Treatment 5, T. harzianum; Treatment 6, T. brevicompactum 6311. Inoculation amount of Trichoderma strains: 10 ml of Trichoderma spore solution with 1 × 106 spores, inoculation amount of Phytophthora: 10 ml of Phytophthora spore solution with 1 × 105 spores. The sentence has been modified on lines 327-331.

Comments 23. Modify the legend of Figure 4.

Reply 23: Thank you for your reminder. The filaments has been replaced with hyphae.

Comments 24. Others comments have been made in the manuscript.

Reply 24: Thank you for the reminder. All comments have been modified.

Reviewer 3 Report

The article is in line with the current needs of agriculture in terms of the need to reduce the use of chemicals. It is an extensive article that addresses everything related to the identified strains of antagonistic fungi of the genus Trichoderma, but requires more organization both in the methodology and in the results, following an order that cannot be violated. Some remarks are made in the document. The title should be adapted to what is discussed in the article, first the pathogen, which is Phytophthora capsici, and then the antagonist, which is Trichoderma spp. until the species are identified, addressing everything from the detailed morphological identification of each species, to the molecular level with dendrograms of each species that include the accession numbers, to the in vitro confrontation of each species with the pathogen, detailing the time in which the antagonist covered the plate completely or partially (use the scale of Bell et al. 1982), and finally the studies of the AIA, siderophores and the tests in pots and germination.

Figure 1, 2 and 4 have many things in the same figure.

Author Response

Comments 1.The article addresses the topic of Phytophthora capsici, specifically, as a pathogen and several Trichoderma species identified in the study, but the role of a specific specie in the results is never highlighted, so the title should focus on Phytophthora capsici and the role of Trichoderma spp, as an antagonist.

Reply 1: Thank you for your suggestion. I have changed the title to Trichoderma brevicompactum 6311: Prevention and control of Phytophthora capsici and its growth-promoting effect

Comments 2.The abstract is not complete, it lacks a discussion of the methodology and the authors do not state the objective of the work. Reference 3 in the introduction is too old to cite statistical data, it should be changed to a citation from 2024. The introduction is disorganized, the authors talk about various species of Trichoderma and how they control pathogenic fungi without a connection and the paragraph is extraordinarily long. In addition, they do not say anything about T. brevicompactum which is the species of the article. The last paragraph should state the objective and they talk about what they will do and how they will do it, that is not part of an introduction. The paragraph is noted in the document.

Reply 2: Thank you for your suggestion. We added a related article published in 2024; We previously summarized that Trichoderma strains can inhibit the growth of Phytophthora capsici, improve plant resistance to pathogens, promote plant growth, and regulate soil microbial structure. It has been successfully developed as a commercial biological control agent; At the same time, we added the biological control cases of T. brevicompactum on plant diseases in lines 80-83, and cited relevant literatures; In the last paragraph of the introduction, we added our goal to provide the basis for the effective control of Pepper Blight and promote the sustainable production of pepper; In addition, we have modified the matters you indicated in the document.

Comments 3.The study is extremely extensive, many things are mixed, so a lot of organization is needed to adequately explain what was done and how it was done. In the methodology where they do the inhibition studies, the strains used for this are never mentioned. The experimental method where they sow the culture and inoculate the Trichoderma strains and the pathogen does not make sense of what they did or why they used the quantities in mL of the antagonist they used. First they talk about treatments 2, 3 and 5 and then 2, 3 and 4, assuming that these treatments are the ones mentioned in the Potting experiment. On the other hand, they state that they do this after 90 days of cultivation and this is a big mistake, you have to inoculate taking into account the pathogen cycle and obviously before the culture finishes its cycle. In this same experiment, the authors state that they collect roots after the fifth day of inoculation and evaluate the severity of the disease taking into account lateral branches, stems and the state of wilting of the plant. They do not mention the fruits, which are edible and are very damaged. The methodology lacks a lot of explanation. Comparison of antagonistic properties between strains: The authors explain a methodology in detail with poor writing. I imagine that this methodology has already been published and the author should be mentioned. In addition, the purpose of what was done is not known.

Reply 3: Thank you for your suggestion. We have added six Trichoderma strains to the inhibition research method. The spore concentrations of Trichoderma strains and Phytophthora capsici were determined when we used the antagonists in the pot experiment. At the same time, in order to avoid ambiguity, we improved the sentences (188-190 lines) in the pot experiment. The goal of our pot experiment is to study the short-term effect of Trichoderma inoculation on Phytophthora capsici disease. The root system of pepper will inevitably be damaged during the transcriptome experiment, so there is no way to count the yield of pepper fruit in this experiment; The reason why we did not mention the author is that we introduced the test method in detail.

Comments 4.The organization of the results is not correct. Results with six antagonistic strains are mentioned, however in a large part of the text they are mentioned as Trichoderma spp., which means that the species is not known, but neither is the race mentioned and figure 1 has a large mix of results from in vitro confrontations, which is what should be shown, to morphology of some of the Trichoderma races and dendrograms. This figure should be limited to showing the results of the confrontations, specifying the day of the result shown, whether it was at 72 hours, 5th day, 6th, etc. Throughout the text, Trichoderma is called biocontrol and in reality it is an antagonistic fungus. In this section, the morphological identification should be separated, then the molecular and finally the in vitro confrontation, because first you have to know what you have in order to then perform the confrontation. It is necessary to show the photos of the morphology of each strain and the dendrograms with the molecular identification and the accession numbers in the NCBI. The names of the species identified are never used in results, authors always use the numbers until discussion so it is very confusing.

Reply 4: Thank you for your suggestion. We call these Trichoderma strains in most words because these six strains of Trichoderma include a variety of Trichoderma. To avoid ambiguity, we have modified the known Trichoderma strains to Trichoderma strains; At the same time, we added the double culture test to the legend in Figure 1 as the confrontation result displayed on day 7; We realize that numbering is easy to be confused, so we modify and correct it in the full text. At the same time, we split Figure 1 into existing figures 1 and 2 to express logic clearly. Morphological and molecular information of six Trichoderma strains In S1-S2; With regard to the molecular identification of strains, I answered the 18th question of the second reviewer, mainly to express that there have been a large number of studies on the identification of Trichoderma strains only using its gene, and the journals published by these studies represent a professional and high level. We are also willing to use more specific primers to supplement Trichoderma identification, but it requires more test time to ensure the identification work.

Comments 5. The article is in line with the current needs of agriculture in terms of the need to reduce the use of chemicals. It is an extensive article that addresses everything related to the identified strains of antagonistic fungi of the genus Trichoderma, but requires more organization both in the methodology and in the results, following an order that cannot be violated. Some remarks are made in the document. The title should be adapted to what is discussed in the article, first the pathogen, which is Phytophthora capsici, and then the antagonist, which is Trichoderma spp. until the species are identified, addressing everything from the detailed morphological identification of each species, to the molecular level with dendrograms of each species that include the accession numbers, to the in vitro confrontation of each species with the pathogen, detailing the time in which the antagonist covered the plate completely or partially (use the scale of Bell et al. 1982), and finally the studies of the AIA, siderophores and the tests in pots and germination.

Reply 5: Thank you for your suggestion. We believe that the logical order you mentioned is a very important focus, and we have revised the topic to be more suitable for the discussion T. brevicompactum 6311: Prevention and control of Phytophthora capsici and its growth-promoting effect; We isolated 6 highly efficient biocontrol strains through a dual culture experiment, and then performed morphological and molecular identification on them; We will present the morphological and molecular biology results of the remaining strains of Trichoderma in the attached figures S1-S2; We chose to measure and calculate the corresponding results on the seventh day by referring to the results of the dual culture experiment of Trichoderma and Phytophthora capsici in previous literature. We have read and understood Bell et al.'s 1982 scale, but we believe that using inhibition rate can better express the results of the dual culture experiment. Thank you for your feedback。

Comments 6. First person cannot be use in papers.

Reply 6: Thank you for your suggestion. We will modify the first person in the abstract and introduction sections to the third person

Comments 7. The writing needs to be improved and punctuation marks need to be used.

Reply 7: Thank you for your suggestion. We have added a period in line 102-105.

Comments 8. Wich treatments? In this case is necessary to give control an advantage of at least 72 and then put the antagonist

Reply 8: Thank you for your feedback. I have read some relevant literature, such as Xia et al.'s Characterization and antagonistic potentials of selected Rhizosphere Trichoderma species against some Fusarium species. The article shows that in a dual culture experiment, both biocontrol and pathogenic bacteria were inoculated and tested simultaneously.

Comments 9. This is not clear. What I understand is that after 90 days of sowing the seeds, 6 and 10 mL of Trichoderma 2213 + P. capsici, Trichoderma 6311 + P. capsici and Trichoderma 2213 were applied. Is that so? And if so, why 6 and 10 mL?

Reply 9: Thank you for your suggestion. What we want to express is to add 10ml spore solution to each of processing 2, processing 3, processing 5, and processing 6. To avoid ambiguity, we have modified this sentence in lines 188-190.

Comments 10. Which ones?

Reply 10: Thank you for your suggestion. What I want to express is to conduct experiments on 6 strains of Trichoderma, with three replicates for each treatment group. I have added this information to the first sentence.

Round 2

Reviewer 1 Report

In the previous review, the mentioned topic on below was not replied: 

"However, the manuscript has weaknesses in the detection of Trichoderma species. They used only ITS region primers. For accurate species diagnosis, they must apply a second primer pair that amplifies different genes in Trichoderma and perform blastn analysis."

For Trichoderma species identification, housekeeping genes together with ITS must be sequenced and analyzed by blast analysis. Other hand, you can not mention the species. You identified only the trichoderma genus. 

In the previous review, the mentioned topic on below was not replied: 

"However, the manuscript has weaknesses in the detection of Trichoderma species. They used only ITS region primers. For accurate species diagnosis, they must apply a second primer pair that amplifies different genes in Trichoderma and perform blastn analysis."

For Trichoderma species identification, housekeeping genes together with ITS must be sequenced and analyzed by blast analysis. Other hand, you can not mention the species. You identified only the trichoderma genus. 

The minor corrections indicated at the attached file.

Author Response

Thank you for your suggestion. I have made minor corrections to the PDF text. Below is my response to your comments.

Comments 1: In the previous review, there was no response regarding the use of only ITS region primers for molecular identification of Trichoderma species. At the same time, for the identification of Trichoderma species, it is necessary to sequence the housekeeping genes with ITS. Other hand, you can not mention the species. You identified only the Trichoderma genus.

Reply 1: Thank you for your suggestion. We deeply apologize for forgetting to reply to your suggestion before. Based on your feedback, in order to accurately identify the molecular characteristics of Trichoderma strains, we used the translation extension factor tef1 as a primer in the new manuscript for identification experiments. The specific steps are as follows: using the improved CTAB method described by Tiwari to extract total genomic DNA from mycelium, we used the following primers: EF1-728F (5 '- CAT CGA GAA GTT CGA GAA GG-3') and TEFl rev (5 '- TAC TTG AAG GAA CCC TTA CC-3'). Perform PCR according to Korkom's procedure and directly sequence the purified PCR product. Construct the tef1 sequence of biocontrol bacteria together with the sequences of various strains of Trichoderma and other strains in the NCBI GenBank database, and use neighbor linkage to construct a phylogenetic tree in the MEGA11 program. The results showed that 2213 and 2221 were molecularly identified as T. harzianum; 5111, 6311, and 6321 were identified as T. brevicompactum; and 7111 was identified as T. virens, which is consistent with the results obtained from ITS identification. These two experiments confirmed the reliability of the identification results, indicating that we have determined the species level of wood mold through experiments. We made modifications to the experimental methods section from lines 141 to 143. The modified parts in the results are shown in lines 299-304 and Figure S3 in the attachment.

Reviewer 3 Report

The topic is very relevant and current, as I had already stated in the first review, and the article addresses all the methodology necessary to demonstrate the usefulness and effectiveness of the antagonists under study, from morphology to molecular aspects, in vitro confrontations and the in vivo experiment. The authors made all the suggested changes, so I consider that the article has the scientific quality necessary to be published. All that remains is to review small grammatical aspects, capitalization, in vitro writing in italics.

Nothing

Author Response

Comments 1: The topic is very relevant and current, as I had already stated in the first review, and the article addresses all the methodology necessary to demonstrate the usefulness and effectiveness of the antagonists under study, from morphology to molecular aspects, in vitro confrontations and the in vivo experiment. The authors made all the suggested changes, so I consider that the article has the scientific quality necessary to be published. All that remains is to review small grammatical aspects, capitalization, in vitro writing in italics.

Reply 1: Thank you for your suggestion, and we also appreciate your recognition of our response. We have thoroughly checked and revised all grammar aspects, capitalization, and italic writing issues throughout the entire text. Thank you again for your suggestions.

Round 3

Reviewer 1 Report

The authors made all necessary corrections to my questions. 

Line 142: Trichoderma has to be italics

The authors made all necessary corrections to my questions. 

Line 142: Trichoderma has to be italics